# Successful Rehabilitation and Release of a Korean Water Deer (*Hydropotes inermis argyropus*) After a Femoral Head Ostectomy (FHO)

**DOI:** 10.3390/ani15142148

**Published:** 2025-07-21

**Authors:** Sohwon Bae, Minjae Jo, Woojin Shin, Chea-Un Cho, Son-Il Pak, Sangjin Ahn

**Affiliations:** 1College of Veterinary Medicine, Institute of Veterinary Science, Kangwon National University, Chuncheon 24341, Republic of Korea; sowonico@gmail.com (S.B.); apffltk1@gmail.com (M.J.); paksi@kangwon.ac.kr (S.-I.P.); 2Gangwon Wildlife Medical Rescue Center, Chuncheon 24341, Republic of Korea; disk8126@gmail.com; 3Yanggu Long-Tailed Goral and Musk Deer Center, Yanggu 24506, Republic of Korea; goralmusk71@korea.kr

**Keywords:** water deer, femoral head ostectomy (FHO), hip luxation, wildlife rehabilitation, GPS tracking, conservation medicine

## Abstract

A young male water deer was rescued after a vehicle collision and diagnosed with hip luxation. To relieve pain and restore mobility, femoral head ostectomy surgery was performed. The water deer subsequently underwent a structured rehabilitation program and was later released into its natural habitat equipped with a GPS tracking collar. Movement data confirmed that the water deer successfully adapted to its environment and regained normal mobility. This case highlights that surgical treatment, combined with rehabilitation and GPS-based monitoring, can be an effective strategy for managing orthopedic injuries in wild animals and supporting their successful return to the wild.

## 1. Introduction

Roadkill and vehicle collisions are significant causes of injury and mortality in wildlife worldwide, particularly in regions with expanding road networks and fragmented habitats [1,2]. In Korea, the water deer (*Hydropotes inermis argyropus*) has been frequently reported as a representative species involved in wildlife-vehicle collisions, often resulting in serious injuries or death [3]. The water deer is the most frequently encountered wild mammal species across South Korea, with roadkill incidents reported extensively along national highways, regional roads, and local routes [4]. Between 2004 and 2019, highway mortality records documented 36,863 roadkill events, of which 28,045 cases (76.1%) involved water deer, highlighting their particular vulnerability to vehicular collisions [5]. These accidents often result in various traumatic injuries such as thermal skin damage, fractures of the limbs or spine, hip dislocations, pelvic trauma, and cranial injuries. Such trauma has been consistently cited in the literature as a leading cause of mortality in cervid species [6,7,8,9]. Although widespread across most regions of Korea, the water deer is listed as a globally threatened species, categorized as vulnerable on the International Union for Conservation of Nature Red List, due to its restricted range and declining populations in China and other locations [10]. These small and solitary ungulates are especially vulnerable due to their crepuscular activity patterns and tendency to cross roads that intersect their habitat ranges [11].

Among the various types of trauma resulting from vehicle collision, musculoskeletal injuries, including long bone fractures and joint dislocations, are common [12]. Coxofemoral luxation, or the dislocation of the hip joint, is a severe injury that compromises the animal’s ability to walk, forage, and escape predators, thus posing a direct threat to its survival in the wild [13]. In clinical veterinary practice, a femoral head ostectomy (FHO) is a well-established surgical procedure for managing intractable hip luxation in companion animals [14,15,16]. However, its application in wild ungulates remains relatively undocumented.

This case study presents a successful FHO procedure performed on a water deer following vehicle collision, its subsequent rehabilitation process, and its post-release monitoring using GPS tracking technology. The integration of surgical intervention, rehabilitation, and ecological monitoring establishes an integrated clinical and ecological management framework in wildlife medicine, contributing to informed conservation strategies and enhanced animal welfare outcomes.

## 2. Case Presentation

### 2.1. Rescue and Initial Assessment

A young adult male water deer, weighing 19 kg (the body weight of Korean water deer typically ranges from 16 to 21 kg) was found following a vehicle collision and was transported to the Kangwon National University Wildlife Medical Rescue Center [17]. Upon arrival, the water deer exhibited severe right hindlimb lameness with a non-weight bearing posture. Physical examination revealed pain and swelling around the hip joint, while radiographic imaging confirmed unilateral coxofemoral luxation without acetabular or femoral head fractures (Figure 1).

### 2.2. Surgical Intervention

Due to the severity of the luxation and the unlikelihood of successful closed reduction, an FHO was chosen as the preferred surgical approach (Figure 2). The procedure was performed under general anesthesia using a combination of xylazine HCl (1.1 mg/kg IM) for induction, followed by isoflurane maintenance. The femoral head was surgically excised to eliminate bone-on-bone contact and reduce pain, promoting the formation of a pseudoarthrosis. Cefazolin (10 mg/kg IV), famotidine (0.5 mg/kg IV), and tramadol HCl (2 mg/kg IV) were administered twice daily for seven days to prevent infection, manage pain, and protect the gastrointestinal mucosa. Meloxicam (0.5 mg/kg SC) was administered once daily for the same period to control inflammation. Due to effective infection and pain control, skin sutures were removed two weeks postoperatively, and the animal demonstrated a stable gait without complications.

### 2.3. Postoperative Rehabilitation

Postoperatively, the water deer was housed in a restricted indoor enclosure for two weeks to prevent excessive movement and support early recovery. A structured rehabilitation program was implemented during this period, including a once-daily passive range of motion exercises, gradual weight-bearing activities, and controlled movement to strengthen the affected limb (Figure 3A). The water deer progressively regained mobility and was able to place an even weight on all four limbs. Subsequently, the animal was transferred to an outdoor semi-natural enclosure for approximately six months, where it continued rehabilitation under more natural conditions. This phase allowed for the further development of muscle strength, coordination, and normal locomotor function (Figure 3B). By the end of the rehabilitation period, the water deer exhibited full weight-bearing capacity and coordinated movement.

### 2.4. Post-Release Monitoring: GPS Tracking and Home Range Analysis

After a successful rehabilitation period, the water deer was fitted with a GPS tracking collar and released into a suitable natural habitat. Considering research indicating that transmitter weight can affect animal behavior, the collar was designed to weigh less than 5% of the water deer’s body weight, to minimize discomfort (Figure 4A,B) [18].

GPS data were collected over a 10-month post-release monitoring period to assess movement patterns and habitat utilization. Home range analysis was conducted using the Minimum Convex Polygon (MCP) method with the Home Range Tools 2 extension in ArcGIS 10.3 (ESRI Inc., Rediands, CA). MCP analysis was performed at both the 95% and 50% levels to delineate the overall home range and the core habitat, respectively (Figure 4C).

The overall home range (MCP 95%) refers to the outer boundary encompassing 95% of the animal’s GPS locations, indicating the general area used for daily activities such as foraging, resting, and movement. In contrast, the core habitat (MCP 50%) represents the central area most frequently used by the animal, often indicating preferred or high-value habitats essential for survival [19].

MCP 95% (overall home range): 8.03 km^2^;MCP 50% (core habitat): 6.967 km^2^.

The GPS data indicated that the water deer successfully adapted to its environment, exhibiting movement patterns comparable to those of healthy individuals. No abnormal sedentary behavior or indications of re-injury were observed during the monitoring period.

## 3. Discussion

This case represents the first reported instance in Korea of a water deer successfully treated with an FHO, followed by structured rehabilitation and GPS-based post-release monitoring. Although the surgical procedure performed in this case was classified as an FHO, radiographic findings indicate that the excision may have been limited to the femoral head, with the partial preservation of the femoral neck. This raises the possibility that the successful clinical outcome was not solely attributable to the surgical intervention but rather to a combination of effective postoperative medical management, including pain control and anti-inflammatory therapy, and a structured rehabilitation program.

While FHOs have been widely used in companion animals and some exotic species to manage hip joint injuries that cannot be resolved through conventional reduction techniques, their application to wild ungulates such as water deer had not been previously documented in Korea [13,14,15]. Even though the excision may have been limited to the femoral head, the intervention in this case successfully alleviated joint pain and restored ambulation, enabling reintegration into the wild. The key factors contributing to the positive outcome included early surgical intervention, structured postoperative rehabilitation, and effective post-release monitoring.

The GPS tracking results provided critical evidence that the water deer could establish a normal home range, suggesting a high level of functional recovery. Interestingly, the home range observed in this case (MCP 95%: 8.03 km^2^) was substantially larger than that reported in a previous study on Korean water deer, where the mean home range size was 2.77 km^2^ under the MCP 95% method and 0.34 km^2^ under the kernel 50% method [20]. Differences in season and sex may account for the variation; however, this result suggests that the rehabilitated individual regained a comparable or even enhanced level of mobility and ecological integration.

Furthermore, the successful adaptation of the water deer despite undergoing an invasive orthopedic procedure highlights the potential for expanding the indications of FHOs in wildlife medicine, particularly in species where mobility is essential in survival post-release. This case supports the feasibility of applying high-level clinical decision-making and individualized care plans, even in field-based or resource-limited rehabilitation settings. It also demonstrates the value of long-term monitoring tools such as GPS tracking in quantifying the functional post-release outcome of such interventions [21].

The release site, a protected natural area with minimal human disturbance, ample vegetation cover, and abundant forage resources, likely contributed positively to the water deer’s successful post-release adaptation. Ensuring habitat suitability at the release location is a critical factor in wildlife rehabilitation success, particularly for species with strong site fidelity or dietary specificity [22].

Additionally, long-term monitoring tools such as GPS tracking play a key role in evaluating post-release outcomes beyond simple survival [23]. Conservation interventions must be assessed not only by survival but also by behavioral and ecological integration [24]. Similarly, release protocols tailored to the species’ ecological needs significantly improve post-release success in mammalian wildlife [24]. This case exemplifies the importance of matching clinical recovery with ecological competence when planning for reintroduction.

Previous studies have shown that assessing post-release success is crucial in wildlife rehabilitation efforts, as survival alone does not indicate full recovery or adaptation [24]. This study reinforces the value of integrating wildlife medicine, rehabilitation science, and ecological monitoring to improve conservation outcomes. As such, it serves as a pioneering model, demonstrating that advanced veterinary surgical techniques can be applied effectively to native wild ungulates and scientifically evaluating their outcomes through ecological tracking methods.

## 4. Conclusions

This case report demonstrates that an FHO represents a feasible and effective surgical approach to managing hip luxation in water deer. These findings highlight the importance of multidisciplinary approaches in wildlife medicine and conservation, emphasizing the need for continued research and monitoring to optimize rehabilitation protocols for injured wild animals.

## Figures and Tables

**Figure 1 animals-15-02148-f001:**
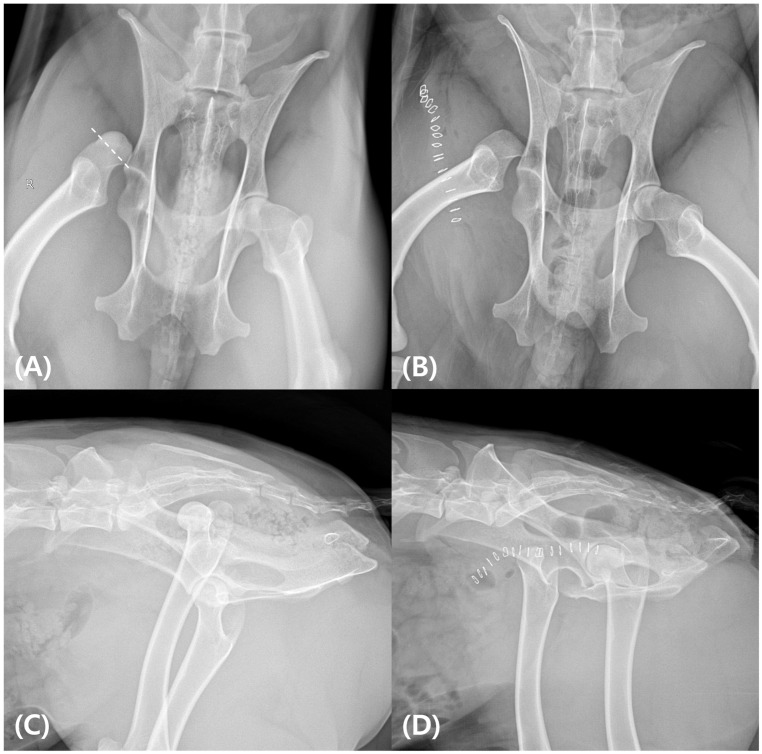
Radiographic findings of the hip joint of the water deer. (**A**) Preoperative ventrodorsal radiograph showing unilateral coxofemoral luxation without evidence of femoral head or acetabular fracture. The expected resection area for the femoral head ostectomy is indicated with a white dotted line. (**B**) Postoperative ventrodorsal view following femoral head excision. (**C**) Preoperative lateral view showing the dislocation of the right femoral head. (**D**) Postoperative lateral view confirming the complete removal of the right femoral head.

**Figure 2 animals-15-02148-f002:**
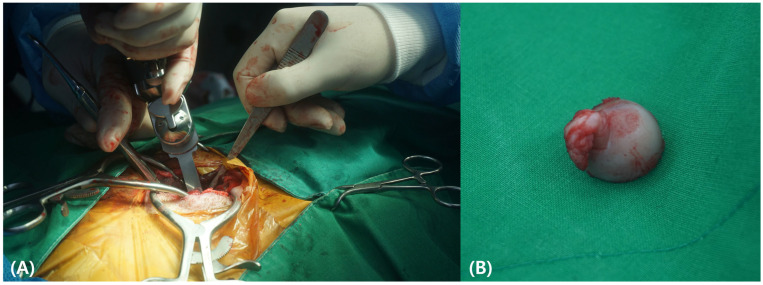
Intraoperative photographs of the femoral head ostectomy procedure. (**A**) Surgical exposure of the coxofemoral joint during femoral head and neck excision. (**B**) Excised femoral head and neck confirming complete removal to facilitate pseudoarthrosis formation.

**Figure 3 animals-15-02148-f003:**
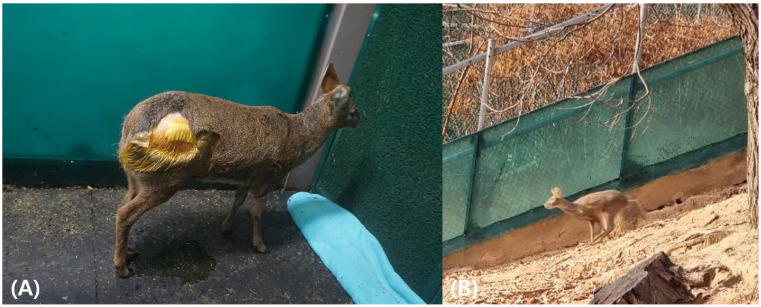
Postoperative rehabilitation of the water deer. (**A**) Indoor enclosure used during early postoperative care, where controlled physiotherapy was provided. (**B**) Outdoor semi-natural enclosure used for gradual reconditioning and functional rehabilitation.

**Figure 4 animals-15-02148-f004:**
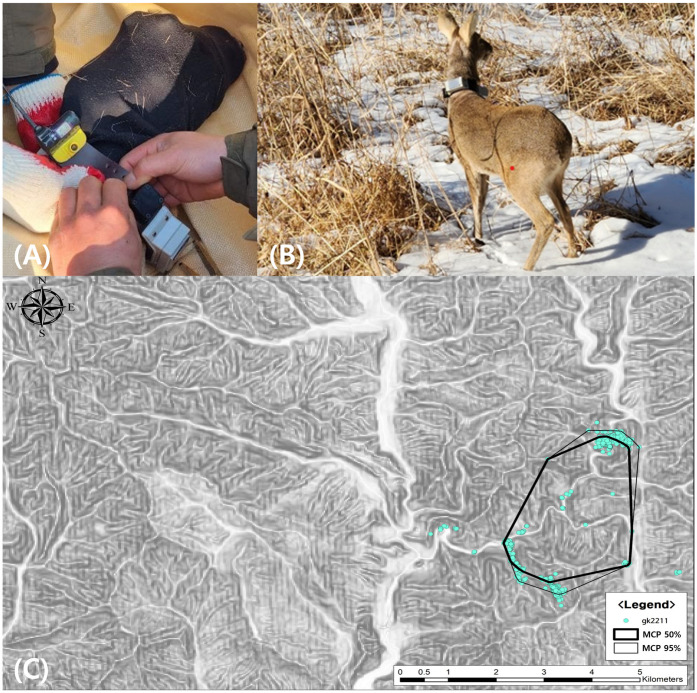
Post-release monitoring using GPS tracking: (**A**) fitting of a GPS collar designed to weigh less than 5% of the water deer’s body weight. (**B**) The water deer immediately prior to release into a natural habitat. (**C**) Home range map generated using the Minimum Convex Polygon method (95% and 50%), illustrating the movement and core habitat area of the water deer.

## Data Availability

The original contributions presented in this study are included in the article. Further inquiries can be directed to the corresponding author.

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
