# Peer review of "Successful Rehabilitation and Release of a Korean Water Deer (Hydropotes inermis argyropus) After a Femoral Head Ostectomy (FHO)"

_animals, 2025, doi:10.3390/ani15142148_

Round 1

Reviewer 1 Report

Comments and Suggestions for Authors

Very interesting case. Just some minor comments mainly regarding figures.

Author Response

Response to Reviewers

We sincerely thank the reviewers for their thoughtful and constructive feedback. We believe that the comments significantly improved the clarity, accuracy, and overall quality of our manuscript. In response, we have carefully addressed all points raised and revised the manuscript accordingly.

Below, we provide a point-by-point response to each comment, including a description of the modifications made. All changes have been marked in red within the manuscript for your convenience.

Reviewer 1

Comment 1: Fig. 1B: The resection of the femoral head and neck is not clearly visible on this X-ray; a more detailed description, perhaps with labelling of the structures and where they were removed, would perhaps be advantageous here.

Response 1:

Thank you for your suggestion. To clarify the resection area, I revised Figure 1A by adding a white dotted line indicating the expected region of femoral head excision.

The figure legend was modified as follows:

  • Figure 1. Radiographic findings of the hip joint of the water deer. (A) Preoperative ventrodorsal radiograph showing unilateral coxofemoral luxation without evidence of femoral head or acetabular fracture. The expected resection area for the femoral head ostectomy is indicated with a white dotted line.

Comment 2: Fig. 3A: The legend describes early controlled physiotherapy in an indoor area. However, the picture only shows the animal standing. This seems a little confusing, I would imagine a picture with exercises or therapy. It would be better to write: exercises were carried out in an indoor area (as can be seen here) or something like that. The same for Fig. 3B.

Response 2:

I agree with the reviewer’s observation and have revised the figure legend to clarify that the photographs depict the location where the rehabilitation took place, rather than specific therapeutic activities:

  • Figure 3. Postoperative rehabilitation of the water deer. (A) Indoor enclosure used during early postoperative care, where controlled physiotherapy was provided. (B) Outdoor semi-natural enclosure used for gradual reconditioning and functional rehabilitation.

Comment 3: Line 112 or point: 2.4: A more detailed description of the ‘Overall home range’ and the ‘Core habitat’ would be helpful here, as it is sometimes difficult for non-insiders to understand.

Response 3:

Thank you for pointing this out. I added the following explanation to clarify the definition of MCP 95% and 50%, along with an appropriate reference (now listed as Reference 19):

  • The overall home range (MCP 95%) refers to the outer boundary encompassing 95% of the animal’s GPS locations, indicating the general area used for daily activities such as foraging, resting, and movement. In contrast, the core habitat (MCP 50%) represents the central area most frequently used by the animal, often indicating preferred or high-value habitats essential for survival [19].

We thank both reviewers once again for their valuable input. The revisions made in response to your comments have strengthened the clarify and scientific value of our manuscript. We hope the revised version meets your expectations and look forward to your further feedback.

Sincerely,

Sangjin Ahn

On behalf of all co-authors

Reviewer 2 Report

Comments and Suggestions for Authors

Thank you for the work you have done to put together this case report. There are several areas which I have outlined below that I would recommend addressing to strengthen your manuscript: 

The term “ostectomy” is more appropriate than “osteotomy” in the context of this procedure and should be used consistently throughout the manuscript.  

Including the average weight of the water deer would be helpful for readers unfamiliar with the species. Size often influences the decision-making and expected outcomes of FHNO procedures in other species and may be relevant to prognosis in this case.  

Based on the provided images, it appears to me that a complete femoral head and neck ostectomy (FHNO) was not performed. The femoral neck does not appear to have been excised, only the femoral head at the level of the capital epiphysis. Clarification here is important as it raises the possibility that clinical improvement in this case may be primarily due to the medical management and rehabilitation utilized, rather than the surgical intervention itself. 

The Case Presentation section would benefit from additional information on postoperative management. Please clarify the duration and frequency of antimicrobial, H2 receptor antagonist, NSAID, and analgesic administration. Were these given only perioperatively or continued postoperatively? Additional details on rehabilitation are also requested. For example, how frequently was passive range of motion performed, how long was the animal confined to the more restricted enclosure versus the outdoor enclosure, and what was the total duration of care prior to release?  

Please specify the length of time the animal was monitored upon release before performing MCP analysis to better contextualize the findings. 

Author Response

Response to Reviewers

We sincerely thank the reviewers for their thoughtful and constructive feedback. We believe that the comments significantly improved the clarity, accuracy, and overall quality of our manuscript. In response, we have carefully addressed all points raised and revised the manuscript accordingly.

Below, we provide a point-by-point response to each comment, including a description of the modifications made. All changes have been marked in red within the manuscript for your convenience.

Reviewer 2

Comment 1: The term “ostectomy” is more appropriate than “osteotomy” in the context of this procedure and should be used consistently throughout the manuscript.  

Response 1:

I appreciate this clarification and have replaced all instances of “osteotomy” with “ostectomy” throughout the manuscript. The title has also been revised to reflect this correction. Furthermore, we added an additional reference on femoral head ostectomy (now Reference 16).

Comment 2: Including the average weight of the water deer would be helpful for readers unfamiliar with the species. Size often influences the decision-making and expected outcomes of FHNO procedures in other species and may be relevant to prognosis in this case.  

Response 2:

Thank you for the suggestion. We added the average body weight range of Korean water deer based on the Mammals of Korea reference (now Reference 17):

  • A young adult male water deer, weighing 19 kg (body weight of Korean water deer typically ranges 16 to 21 kg) was found following a vehicle collision and was transported to the Kangwon National University Wildlife Medical Rescue Center [17].

Comment 3: Based on the provided images, it appears to me that a complete femoral head and neck ostectomy (FHNO) was not performed. The femoral neck does not appear to have been excised, only the femoral head at the level of the capital epiphysis. Clarification here is important as it raises the possibility that clinical improvement in this case may be primarily due to the medical management and rehabilitation utilized, rather than the surgical intervention itself. 

Response 3:

We agree with this observation. We addressed this point in the Discussion section by clarifying that the excision may have been limited to the femoral head, and by acknowledging that the favorable outcome may have resulted from a combination of surgical, medical and rehabilitative factors. We revised Paragraphs 1 and 2 of the Discussion as follows:

  • Paragraph 1 of Discussion: This case represents the first reported instance in Korea of a water deer successfully treated with FHO, followed by structured rehabilitation and GPS-based post-release monitoring. Although the surgical procedure performed in this case was classified as a FHO, radiographic findings indicate that the excision may have been limited to the femoral head, with partial preservation of the femoral neck. This raises the possibility that the successful clinical outcome was not solely attributable to the surgical intervention, but rather to a combination of effective postoperative medical management, including pain control and anti-inflammatory therapy, and a structured rehabilitation program.
  • Paragraph 2 of Discussion: While FHO has been widely used in companion animals and some exotic species to manage hip joint injuries that cannot be resolved through conventional reduction techniques, its application to wild ungulates such as water deer has been undocumented in Korea [13, 14, 15]. Even though the excision may have been limited to the femoral head, the intervention in this case successfully alleviated joint pain and restored ambulation, enabling reintegration into the wild. The key factors contributing to the positive outcome included early surgical intervention, structured postoperative rehabilitation, and effective post-release monitoring.

Comment 4: The Case Presentation section would benefit from additional information on postoperative management. Please clarify the duration and frequency of antimicrobial, H2 receptor antagonist, NSAID, and analgesic administration. Were these given only perioperatively or continued postoperatively?

Response 4:

We added the following description to the Case Presentation section to address this question:

  • Cefazolin (10 ㎎/㎏ IV), famotidine (0.5 ㎎/㎏ IV), and tramadol HCl (2 ㎎/㎏ IV) were administered twice daily for seven days to prevent infection, manage pain, and protect the gastrointestinal mucosa. Meloxicam (0.5 ㎎/㎏ SC) was administered once daily for the same period to control inflammation. Due to effective infection and pain control, skin sutures were removed two weeks postoperatively, and the animal demonstrated stable gait without complications.

Comment 5: Additional details on rehabilitation are also requested. For example, how frequently was passive range of motion performed, how long was the animal confined to the more restricted enclosure versus the outdoor enclosure, and what was the total duration of care prior to release?  

Response 5:

We have expanded the 2.3 Postoperative Rehabilitation section as follows:

(restricted enclosure: two weeks / outdoor enclosure: six months)

  • Postoperatively, the water deer was housed in a restricted indoor enclosure for two weeks to prevent excessive movement and support early recovery. A structured rehabilitation program was implemented during this period, including once-daily passive range of motion exercises, gradual weight-bearing activities, and controlled movement to strengthen the affected limb (Figure 3A). The water deer progressively regained mobility and achieved even weight on all four limbs. Subsequently, the animal was transferred to an outdoor semi-natural enclosure for approximately six months, where it continued rehabilitation under more natural conditions. This phase allowed for further development of muscle strength, coordination, and normal locomotor function (Figure 3B). By the end of the rehabilitation period, the water deer exhibited full weight-bearing capacity and coordinated movement.

Comment 6: Please specify the length of time the animal was monitored upon release before performing MCP analysis to better contextualize the findings. 

Response 6:

We clarified the duration of post-release monitoring in Section 2.4:

  • GPS data were collected over a 10-month post-release monitoring period to assess movement patterns and habitat utilization.

We thank both reviewers once again for their valuable input. The revisions made in response to your comments have strengthened the clarify and scientific value of our manuscript. We hope the revised version meets your expectations and look forward to your further feedback.

Sincerely,

Sangjin Ahn

On behalf of all co-authors

Round 2

Reviewer 2 Report

Comments and Suggestions for Authors

Thank you for your revised manuscript.